# A Novel, Energy-Efficient Smart Speed Adaptation Based on the Gini Coefficient in Autonomous Mobile Robots

**Gürkan Gürgöze** [1,*] **and İbrahim Türkoğlu** [2]

1 Department of Software Engineering, Institute of Science, Firat University, 23119 Elazig, Turkey
2 Department of Software Engineering, Faculty of Technology, Firat University, 23119 Elazig, Turkey
* Correspondence: gurkangurgoze@gmail.com; Tel.: +90-5053956815

**Abstract:** Using energy efficiently is an important parameter in mobile robots. The majority of the energy consumption takes place in the motors. As such, past studies have investigated how to reduce the usage time of motors. Although the relationship between task energy and speed energy is considered in these studies, the qualification of the task, the amount of energy used, and the speed relation have not been taken into account as a whole. Parameters that affect each other in determining the speed profile, such as the criteria by which energy saving is determined, the maximum speed limit, acceleration, the load, and the ground relation, have not been taken into account holistically. Another research focus concerns the need to distribute energy in a balanced manner, in accordance with the qualification of the task, and to ensure the movement occurs at the optimum speed. In this study, a new dynamic (online) intelligent speed and acceleration adaptation method, based on the task structure and energy balance, was developed for a specific path that overcomes the shortcomings of existing models. The Gini coefficient was used for the balanced distribution of energy. Sharp speed changes were prevented with the remaining path and the balanced distribution of the remaining energy. The current model is compared with the trapezoidal speed profile structure.

**Keywords:** speed profile; energy balance; task speed relationship; acceleration profile

## 1. Introduction

Mobile robots have limited energy resources. The efficient use of energy has emerged as an effective solution to this problem. At this point, many energy-efficient task and motion planning (short path planning, path smoothing, speed profiles, etc.) strategies have been developed [1–3]. In both types of studies, the focus is on reducing the motor's running time and the power it consumes during travel in the mobile robot system. This is because, in these systems, the most consumption takes place in the motors. In terms of energy efficiency, task and short-path studies are not sufficient solutions on their own. It is also necessary to consider the power consumption during travel [4,5]. The power consumption of the motors during travel relies heavily on a correct steering angle, speed and acceleration. In mission and motion planning studies, energy-efficient speed optimizations are neglected in terms of task durability and stability, or they are not considered as an effective parameter [6–8].

Current energy-efficient speed profile studies generally take into account the structure of the road, but not the structure, durability, and stability of the task. Energy-efficient speed profiles are based on distance and time. Straight and curved segments of the road play an important role in determining speed. Initially, a trapezoidal speed profile was used in applications [5,9]. However, Wang et al. revealed that this method is not optimal in terms of total energy [10]. Kim and Kim have separately presented energy-efficient speed profiles based on a constant travel time for straight paths and a rotational trajectory. However, a speed profile combining both road sections has not been established [9,11]. Tokekar et al., on the other hand, divided the road into segments and presented energy-efficient combined optimum speed profiles for straight and curved roads. Optimum speed profiles

are calibrated according to the specific load and ground according to the polynomial model. Calibration is performed offline. Recalibration is required for each ground and load. In another study, Tokekar et al. stated that the length of the return path increases if the maximum speed is not limited [12]. The first study considering the relationship between the amount of energy and time was carried out by Broderic et al. [13]. They divided the road into straight and turning segments, and provided acceleration according to the offline-determined gain parameters for each segment of the road. However, they could not achieve a balance of time, energy, and speed [14,15].

The energy efficiency in these studies is a result of the speed profile used. In these studies, the amount of energy use, its limit, and the criteria by which energy saving is determined are not available. The amount of energy remaining in the battery was not taken into account at the point when the mission was completed. However, mobile robots perform their movements according to certain tasks and the amount of power in the battery. The definition of the task and the energy relationship in the battery affect the amount of energy to be used, the endurance of the task, the speed, and acceleration profile. Otherwise, only an energy-efficient operation will be put forward. At the same time, there is no limit to the maximum speed in terms of energy use in the current studies. This can lead to the deterioration of the relationship between energy, speed, and task. At the same time, moving at maximum speed, as in the studies conducted by Broderic et al. and Sharallimo et al., unbalances the distribution of energy between segments of the path to meet the criteria of the task [9–16]. There is no measure to correct this imbalance in the current studies. In addition, acceleration was generally chosen as a constant in all these studies. However, it has been seen in real car studies that creating acceleration and acceleration/deceleration distances according to curved roads or the point where the speed change will occur provides energy-saving and safe travel. A dynamic acceleration profile for online load changes is also often not available [13].

As understood from the studies, it is necessary to consider the task description, the battery status, the amount of energy used, and the motion dynamics (maximum speed limit, acceleration, load, and ground, etc.) in conjunction in order to create the optimum energy-efficient speed profile. Otherwise, energy balance and optimum speed cannot be achieved, as is the case in existing speed profiles.

Here we summarize the remaining parts of the study. In the second part, we describe the methods used. In the third part, the integrated structure of control and motion planning developed with these methods is explained step by step. In the fourth part, simulation studies are carried out and a comparison with the trapezoidal speed profile is performed.

### 1.1. Contributions of the Study

- With the developed method, speed and acceleration profiles were developed in conjunction with the task and battery status.
- The durability of the task was ensured by limiting the maximum speed according to the amount of energy used.
- Smoother speed transitions were achieved with energy-efficient acceleration based on the maximum speed limit on straight roads.
- By using the Gini coefficient, the current energy is used in a way that is balanced according to the task.
- A flexible movement model was created by responding to online changes.

### 1.2. Proposed Energy Efficient Speed—Acceleration Control Model

In general, the developed speed–acceleration control model generates speed and acceleration profiles based on the amount of energy for the straight and curved portions of the existing road. A system block diagram is presented in Figure 1. The proposed method offers a combined solution for straight and curved roads. It consists of two parts. The first part determines the extraction, smoothing, task selection, and energy quantity of the straight and curved sections of the existing road. Thus, the energy selection suitable for

the task is carried out. In addition, a combined path is produced with the path extraction algorithm. In this way, the mobile robot is prevented from unnecessarily changing speed for each curve. This is an important step in conserving energy and preventing unnecessary speed changes.

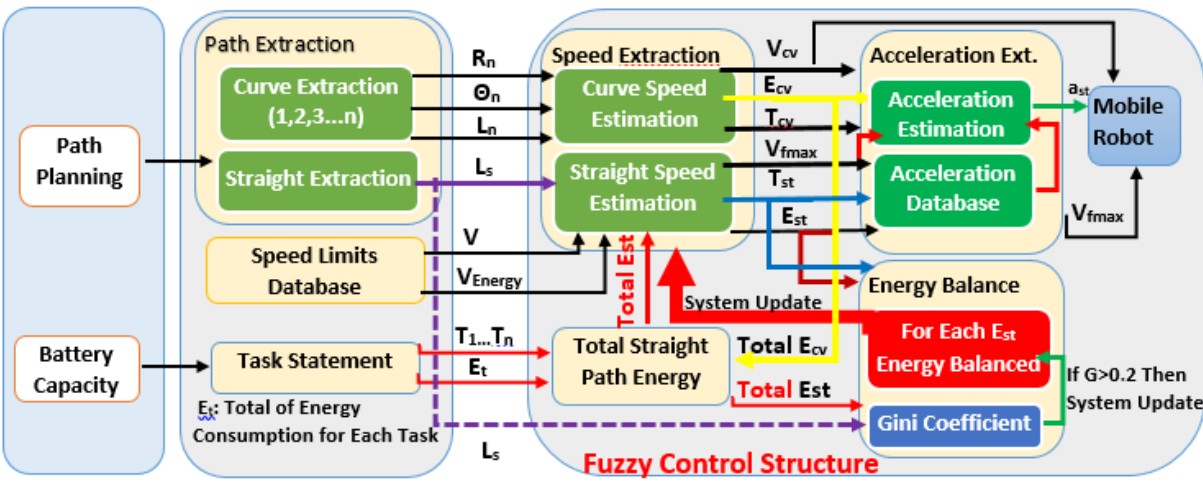

**Figure 1.** System block diagram.

The second part generates linear speed and acceleration values according to the path information, task selection, and energy amount. Thus, the realization of the task is achieved and a more stable system is formed. At the same time, deviations from the road or unnecessary slowdowns are eliminated by providing smoother transitions.

In addition, different accelerations on flat roads, load or ground changes, and other unexpected parameters can cause different energy consumption than expected for each flat road. At this point, the amount of energy to be used for each straight road is recalculated by obtaining the Gini coefficient and the energy distribution imbalance coefficient. Thus, the balanced use of energy is optimized using the Gini coefficient.

The dynamic and energy models in the mobile robot system can provide online responses to the ground friction, slope, and load conditions of the road. Thus, energy changes can be detected. The current speed and acceleration databases in the system were obtained according to different ground and load values. This allows the speed–acceleration control structure to give instant responses in online ground and load changes. Fuzzy logic Mamdani method control was carried out in order to make the whole movement dynamically flexible online. The resulting values are transferred to the motion system [12,15–19].

### 1.3. System Model and Speed Control

The developed speed–acceleration control model creates a hybrid structure, as shown in Figure 2, with the Pure Pursuit path tracking algorithm used in the motion system. It is known that the Pure Pursuit algorithm works with constant linear speed. It generates the expected angular speed values for the next moving point according to the coordinate information obtained from the actual speed values of the system. It follows the path with this angular speed. However, it does not take into account the structure of the road and the amount of energy in its selection of linear speed. When the linear speed value and acceleration of the system change, it can be ensured that the actual speed values reach the expected angular speed values. In this study, a new speed control system is developed by combining the path tracking capability of the Pure Pursuit algorithm and the speed–acceleration information obtained from the developed model for the purposes of energy efficient speed values and safe travel.

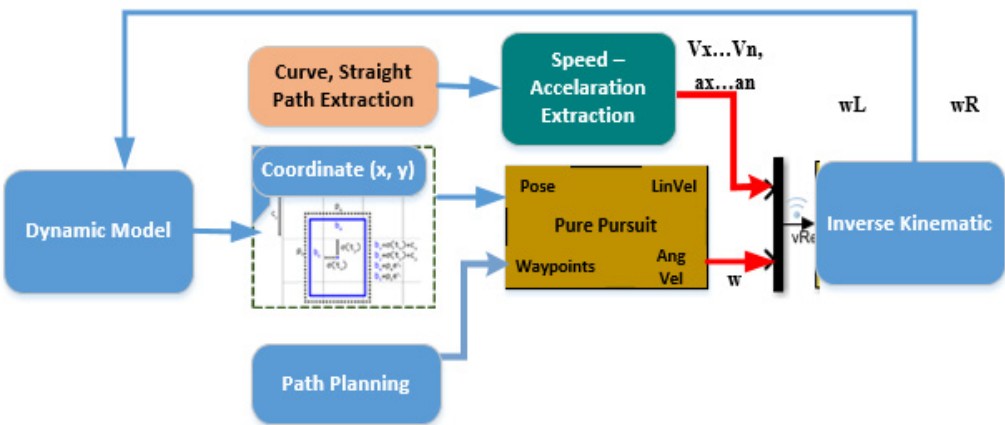

**Figure 2.** Mobile robot system model and speed control structure.

Coordinate information obtained with advanced kinematics in the motion system is transferred to the Pure Pursuit algorithm, and appropriate angular speed values are obtained for the next point. The required linear speed and acceleration information is taken from the developed speed–acceleration control model. This information is transferred to the inverse kinematic model and then to the dynamic structure by obtaining the appropriate right- and left-wheel angular velocities for the next point. It aims to reach these reference speeds with the PID speed controller in the dynamic model. At the same time, sections of the road are obtained with the path estimation algorithm for a correct speed–acceleration selection in this system. Then, the road is smoothed and made suitable for the maneuverability of the mobile robot.

## 2. Materials and Methods

### 2.1. Mobile Robot Model

In this study, we use a new comprehensive mobile robot mathematical model (DC motor model, dynamic and kinematic model), consisting of parameters that are neglected in many studies such as load, slope, kinetic friction, torque saturation, and maximum speed limitation. The DC motor model transfer function is shown in Equation (1):

$$G_s(s) = \frac{\omega(s)}{V(s)} = \frac{K}{(L.s + R_a)(J_m.s + b_m) + K^2} \tag{1}$$

where the application voltage is $V(s)$, the resistance is $R_a$, the inductance is $L$, the inductance constant is $K_e = K_t = K$, the inertia is $J_m$, and the viscous friction is $b_m$.

$$F_M = F_L + F_E + F_Z \tag{2}$$

The net force generated in the $F_M$ motor is the thrust $F_L$, the gravity force of the $F_E$ inclined road, and the $F_Z$ kinematic friction force. (The center of gravity is in the middle of the axis of rotation, and the weight acting on the wheels is m/2).

$$F_L = \frac{m}{2}.a = \frac{m}{2}.\dot{V} = \frac{m}{2}.\dot{w}.r \tag{3}$$

$$F_E = \frac{m}{2}g.\sin\beta \tag{4}$$

The static friction and kinetic friction:

$$F_S = \mu s.F_n.\text{sign}(v) \text{ or } F_K = \mu k.F_n.\text{sign}(v) \tag{5}$$

The viscous friction:

$$F_V = b_z.w \tag{6}$$

The Stribeck effect:

$$F_{ST} = (F_S - F_K).e^{-cs\lceil V \rceil} \tag{7}$$

The friction model:

$$F_{FZ}(V) = F_K + F_{ST} + F_V$$

$$F_Z = \begin{cases} F_{TF} & \text{for } v = 0 \quad \Lambda\ F_{TF} < F_S \\ F_S.\text{sign}(F_{TF}) & \text{for } v = 0 \quad \Lambda\ F_{TF} \geq F_S \\ F_{FZ}(v) & \text{for } v \neq 0 \end{cases} \tag{8}$$

The angle speed is (w), the radius is (r), the speed is (v), the coefficient of kinetic friction is ($\mu_k$) and the coefficient of static friction is ($\mu_S$), the viscous friction is ($b_z$), the coefficient of transition from static to kinetic friction is ($c_s$), and $F_{TF}$ is the force generated in the motor until it reaches the Fs force required for the movement of the mobile robot. $F_{ST}$ is the Stribeck effect, or the total torque generated in the motor during movement. The SIMULINK model is available in Figure 3 [20–23].

$$T_m = T_L + T_E + T_Z = F_L.r + F_E.r + F_Z \tag{9}$$

$$T_m = \frac{m}{2}.r^2.\frac{d^2\theta}{dt^2} + \frac{m}{2}.g.r.\sin\beta + \mu_k.F_N.\text{sign}(v) + (F_S - F_K).e^{-c_s|v|} + b_z.w \tag{10}$$

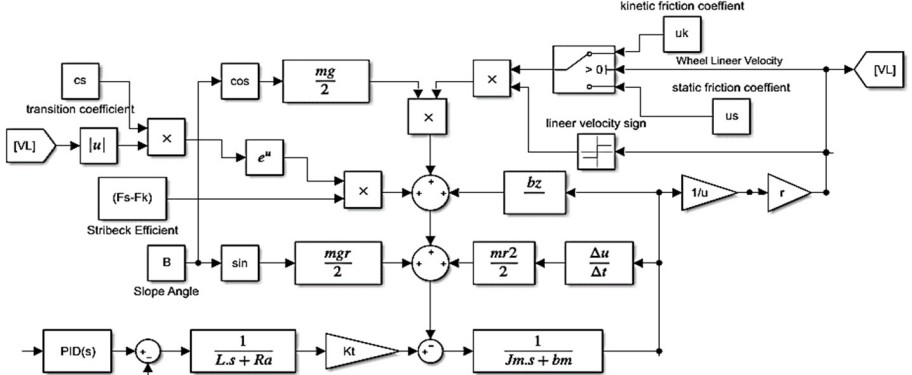

**Figure 3.** Mobile robot SIMULINK Model with friction, slope, and load parameters.

## 2.2. Energy Model

In this study, an advanced energy model that obtains energy consumption according to all internal and external dynamics during the movement of the mobile robot is used. The proposed energy model consists of motors, sensor, and control parts. In this model, there are slope, load, dynamic friction, torque saturation, and maximum speed limitations (curve speed based on centrifugal force, etc.), parameters that are neglected in many studies. The energy consumption of sensors and control structures according to speed and operating status (active, passive, etc.) is also taken into account.

The energy consumption information according to the usage status of the sensors:

$$E_{sensor} = \begin{cases} \dfrac{1}{v_{max}} \int (v.P_{sensor}).dt\ , & v > 0,\ V_{max},\ \text{or}\ \omega_{max} \\ 0, & v \leq 0 \end{cases} \tag{11}$$

The energy consumption of the control part in standby, start-up, and continuous (stable) operation:

$$E_{control} = \begin{cases} E_{standby} = P_{standby}.\Delta t \\ E_{startup} = \int \left( \varphi.\Delta V + \left( \dfrac{t^2}{10} \right) + P_{standby} \right) dt \\ E_{stable} = \int (P_{standby} + t^2) dt \end{cases} \tag{12}$$

$P_{standby}$ is the power consumption at standby, $\varphi$ is the energy factor given to start the controller, and $\Delta v$ is the instantaneous speed change [24].

The energy consumption ($P_k$) in the motor depends on the torque ($T_m$) against the movement and the angular speed ($\omega$).

$$P_k = T_m.\omega \tag{13}$$

$T_m$ was obtained based on friction, slope, and load. If we consider the maximum amount of torque that the motor can produce and the maximum speed it can reach, which is usually neglected in the literature, the energy consumption in the motor is:

$$P_{max} = T_{max}.\omega_{max} \text{ or } P_{max} = F_{max}.V_{max} \tag{14}$$

$$P_k = \begin{cases} P_k, & v < V_{max}, \text{ or } \omega_{max} \\ P_{max}, & v \geq V_{max}, \text{ or } \omega_{max} \end{cases} \tag{15}$$

$$E_s = \int P_k(t) dt \tag{16}$$

Heat energy released by motion:

$$E_h = \int (\epsilon.t^2 + \sigma.v + \lambda.t) dt \tag{17}$$

$$E_h = \int (\epsilon.t^2 + \sigma.v + \lambda.t) dt \tag{18}$$

Here, $\epsilon$ and $\lambda$ are the heat time constant and $\sigma$ is the velocity heat constant.

$$E_{motion} = E_s + E_h \tag{19}$$

Total energy consumed during movement (SIMULINK Model available in Figure 4) [12,14,20,24–27]:

$$E_{total} = E_{sensor} + E_{control} + E_{motion} \tag{20}$$

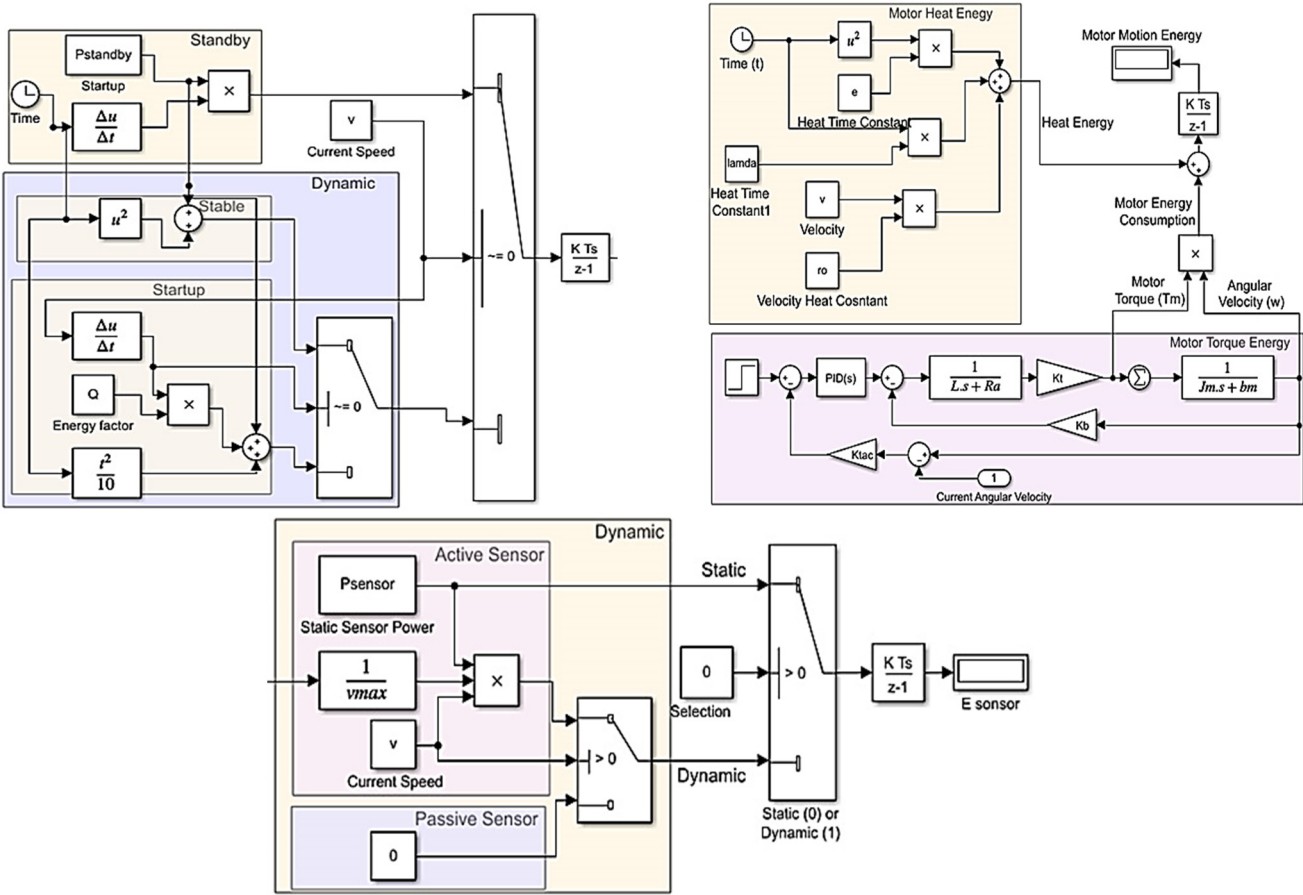

**Figure 4.** Energy consumption SIMULINK Model.

## 2.3. Obtaining Road Segments

The roads consist of two different sections, straight and curved. The path inference method of Li et al., was used to obtain the path segments in Figure 5. At the same time, thanks to this method, softened curved paths suitable for the structure of the mobile robot were created. Then, speed and acceleration profiles were obtained for the acquired road sections.

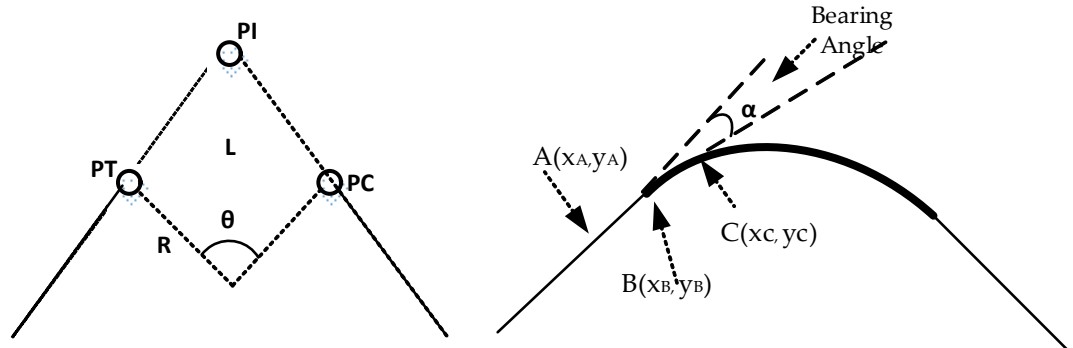

**Figure 5.** Representations of curvature parameters.

The first step in defining the curve of the path is to determine the start and end points of the curve. The determination of these points is achieved by determining whether

the waypoints are part of the curve. To achieve this, the bearing angle ($\alpha$) of each point is calculated.

$$\alpha \cos^{-1}\left(\frac{(x_B-x_A)(x_C-x_B) + (y_B-y_A)(y_C-y_B)}{\sqrt{(x_B-x_A)^2 + (y_B-y_A)^2}.\sqrt{(x_C-x_B)^2 + (y_C-y_B)^2}}\right).\frac{180}{\pi} \tag{21}$$

A threshold value defines whether the current waypoint is part of the curve. In this study, the bed angle was chosen as 1.25. Curves are also defined as simple and compound curves as in Figure 6. The composite curve is created by joining consecutive curves with each other at a certain distance. A length threshold distance is given. Consecutive paths below this threshold value are defined as curves. This allows us to avoid speed variations caused by short-distance sequential curves.

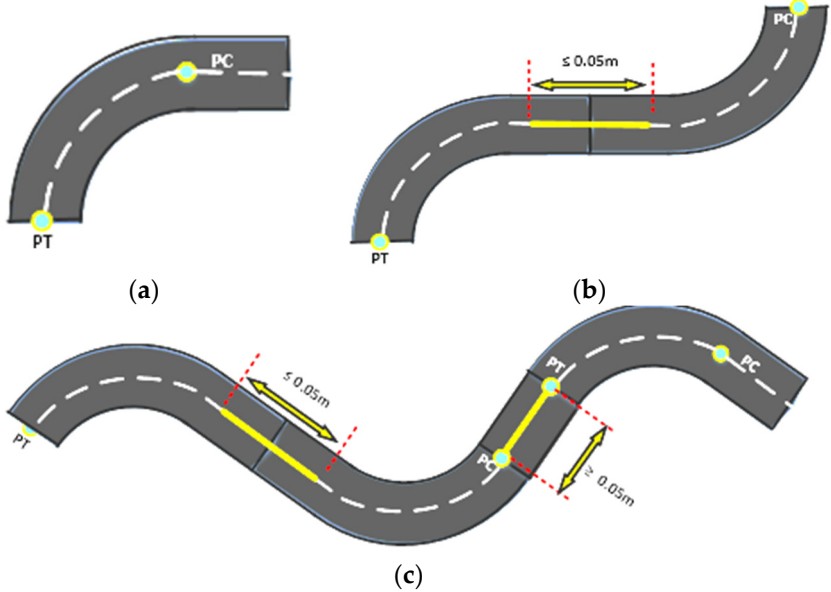

**Figure 6.** Curve classes: (**a**) simple curve, (**b**) compound curve, (**c**) compound and simple curve.

After the start and end points of the curve are determined, the length (L), radius (R), and center angle ($\theta$) information for the curve are found.

$$R = \sqrt{(x_{pc}-x_o)^2 + (y_{pc}-y_o)^2} \tag{22}$$

$$C = \sqrt{(x_{pt}-x_{pc})^2 + (y_{pt}-y_{pc})^2} \tag{23}$$

$$\theta = 2.\sin^{-1}\left(\frac{C}{2R}\right).\frac{180}{\pi} \tag{24}$$

$$L = \frac{\theta.\pi}{180}.R \tag{25}$$

Segments outside the curve are recorded as straight path segments [17–24].

### 2.4. Speed and Energy Profiles of a Curved Road

The speed estimation for the curved parts of the road was obtained by considering the centrifugal force expressed in Equation (28):

$$e = \tan \rho \text{ and } k = 1/r \tag{26}$$

$$v_{cv} = \sqrt{\frac{(e + \mu)g}{k}} \tag{27}$$

where $v_{cv}$ is the curve speed, e is the super height, r is the radius of the curve, $\theta = 5^n \leq 90$ curve angle, and $\mu$ is the ground friction coefficient [17,18].

The amount of angle change in the folds is realized according to the predetermined curve threshold value. As shown in Figure 7, the following information was obtained: 5° angle differences between 0–90° for roads and 0.5 radius steps between r = 0.5–10 m and 10 kg steps between m = 10–50 kg and constant speed profiles ($V_{cv}$), for 2 different soils (stone, asphalt), energy consumption ($E_{cv}$), distance ($X_{cv}$) and time information ($T_{cv}$). The proposed energy model was used for the estimated energy consumption information. The stone and asphalt static friction coefficient $\mu_s$ = 1.0, while the kinetic friction coefficients are stone ground $\mu_k$ = 0.7 and asphalt ground $\mu_k$ = 0.5.

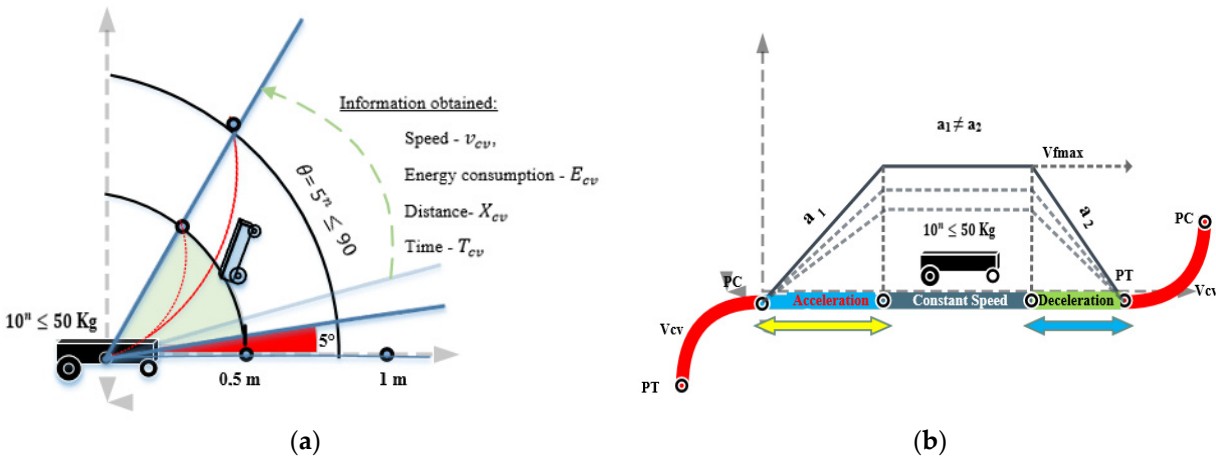

(a)　　　　　　　　　　　　　　　　　　　　　　(b)

**Figure 7.** (**a**) Speed profiling, (**b**) acceleration profiling.

*2.5. Load and Speed Based Acceleration Profile of Straight Path*

Acceleration and load are two important factors that increase energy consumption. Being able to select the acceleration at small values based on the load and increasing the constant speed distance also means reducing the energy consumption. In previous studies, constant acceleration, which is determined offline and at minimum values, has generally been used for straight road segments. Constant acceleration was chosen without considering any other criteria. However, such analyses do not consider whether this selection is suitable for the duration, energy consumption, maximum speed limit, or load parameters of the task during the movement. For this reason, in this study, acceleration profiles were created to select the appropriate acceleration. At the same time, it is ensured that the task criteria are met by the acceleration selection.

The acceleration profile information is obtained by reaching the maximum speed with acceleration values at certain intervals, according to the characteristics of the mobile robot at certain loads and on the ground. Speed change values were selected in the range of 0–2.5 m/s with 0.5 m/s step differences. The acceleration was changed in 0.1 m/s² steps in the range of 0–2 m/s², and the load was changed in 10 kg intervals up to 50 kg. Depending on the acceleration value, energy consumption ($E_{at}$), distance ($X_{at}$), and the time to reach the destination ($T_{at}$) can also be included in the profiles. Contrary to other studies, the acceleration distance of the straight road was obtained based on energy [9–12].

*2.6. Task Definition and Energy Quantities*

In this study, the first step in the energy-oriented selection of the speed and acceleration profiles is to determine the amount of energy that can be used according to the structure of the task and battery status. Thus, a balanced speed and acceleration can be achieved with a

balanced energy distribution. Since the amount of energy and its use depend on the task, task definitions should first be determined.

Tasks:

- Task 1: it is a fixed-duration and fixed-distance task type. There is no energy limit. The maximum speed limit is determined by the amount of energy remaining in the battery.
- Task 2: a fixed-distance and $V_{min}/V_{max}$ fast task type. There is no energy and time limit. It generally has the desired speed profile. Energy is the amount of full or remaining energy in the battery.
- Task 3: a fixed-distance and $V_{min}$ fast task type. There is no energy or time limit.
- Task 4: a fixed-distance- and constant-energy-type task. There is no time limit. The amount of energy to be used is defined.
- Task 5: the type of task with an unknown distance and constant energy. The amount of energy is the full or remaining energy of the battery. The mobile robot performs its movement according to the first exit point, certain charging points, or battery change points. It is preferred for wide-area scanning tasks
- Task 6: the type of task with a constant duration and constant energy. It aims to travel at the maximum speed limit suitable for the time and energy balance. If the battery is full, the movement is performed with $V_{max}$.
- Task 7: a fixed-distance task type with N repetitions. The amount of energy for each repetition is determined by the balanced distribution of the full or remaining amount in the battery. The velocities of the repetitive movement are obtained according to the amount of energy. The movement speed of the full battery is expected to be $V_{max}$, and the remaining battery amount to be $V_{fmax}$.
- Task 8: a fixed-time task type with N repetitions. The amount of energy for each repetition is determined by travel with $V_{max}$ or $V_{fmax}$.

According to the task definitions, the amount of energy is determined according to two basic situations. Here, the fixed energy $E_t$ can be an estimated value, the amount of energy remaining in the battery, or a divided amount for repeated run-ons. The first is the constant-energy state, where the amount of energy to be used is known. With the defined constant energy, $V_{fmax}$ or $V_{min}$, $V_{cv}$ speed profiles and $a_{st}$ straight road acceleration profiles are determined. Traveling at a maximum speed is generally expected in this type of work. However, in some tasks where there is no time and distance information or where it is necessary to go slowly for safety, a minimum speed may be required according to the energy status of the battery.

The second situation is where the amount of energy to be used is obtained depending on the task time ($T_t$), load ($m$), ground friction ($b_z$), distance information ($X_t$), and $V_{max}$ or $V_{min}$ speed limit. It is not known how much energy is required based on the completion status of the task. In such tasks, the state of the battery and the minimum or maximum selection of the speed are especially important.

Depending on the dynamics of the environment and unexpected situations in the given tasks, the energy balance on straight roads may deteriorate. This situation causes the rates to vary more than expected for the task completion criterion.

This means that more energy and time are needed. In order to prevent this, a balanced distribution of the remaining energy during the movement should be achieved, according to the nature of the task. For a balanced distribution of the remaining energy on straight roads, we need to obtain the energy imbalance ratios. The imbalance of the energy distribution can be found using the Gini coefficient.

### 2.7. Energy Balancing Based on the Gini Coefficient

In this study, the Gini coefficient is used to provide a balanced distribution of the energy remaining during the motion between straight road segments. Thus, applications based on time and distance constraints aim for optimum acceleration and energy-efficient movement.

The Gini coefficient was developed by the Italian statistician C. Gini to reveal the income imbalance of countries. It is based on the Lorenz curve. Distribution imbalance is expressed in the range of 0–1. With regard to the distribution of a small number of data variables, its success is especially high. The closer the Gini coefficient is to zero, the fairer it is, and the farther it is, the more unbalanced it is. Some limits have been defined by the United Nations to express income imbalance. A range of 0–0.2 indicates absolute income, 0.2–0.3 indicates the relative average, 0.3–0.4 indicates a reasonable average, 0.4–0.5 indicates significant imbalance, and 0.5 and above indicates income gap. Zhang's approximate trapezoidal field theory in Equation (29) was used to find the Gini Coefficient:

$$G_i = 1 - \frac{1}{n}\left(2\sum_{i=1}^{n-1} w_i + 1\right) \tag{28}$$

To obtain the Gini coefficient, the energy amount of n straight path segments is ordered from smallest to largest. Then, the percentage ratio ($w_i$) of the energy of the i'th straight track segment to the total straight track energy amount is calculated. Finally, the Gini coefficient of each straight road segment is found using Equation (29) [28].

### 2.8. Determining the Speed and Acceleration Profiles

It is well known that the movement environment of mobile robots is full of uncertainties. It is necessary to give an appropriate instant response to these uncertainties. Therefore, in this study, the fuzzy logic Mamdani method was used due to its low processing speed, smooth calculation, simple control, and fast response. Membership functions are assigned as triangle type for input values and singular for output variables. The center of gravity method was used as the clarification method.

In this study, three-state control was carried out with fuzzy logic. The first is online ground and load calibration. Thus, the offline calibration problem in the previous studies by Tokekar, Broderic, Jaramillo, and Serralheiro has been solved. The second is to achieve task-appropriate energy control. Thus, the energy required for movement can be determined and the balanced use of energy is ensured. The third is to determine the acceleration and speed profiles suitable for the energy, instantaneous environment dynamics, and the type of task. Speed and acceleration profiles were obtained with the existing dynamic structure based on load and ground [12–16,29].

The first step for a particular path is to determine the segments of the path offline. Straight paths ($ST_i$) and their lengths ($X_{st}$), and curved paths ($CV_i$) and their radius (R), angle ($\theta$), and length information ($X_{cv}$) are obtained. The second process is to determine the appropriate curve speed and energy and time information with the fuzzy structure, according to the curve information. The fuzzy structure to be used for this consists of four inputs and three outputs. The inputs are R, $\theta$, m, and $b_z$. The outputs are $V_{cv}$, $E_{cv}$, $T_{cv}$. Membership functions and rule table were created according to the intervals and result values of the speed profiles obtained previously.

The fuzzy rule structure is as follows.

Rule 1: $V_{cv}$: IF R $R_l$ and $b_z$ $Bz_m$ THEN $U_1$ $V_{cv}$
Rule 2: $E_{cv}$: IF $\theta$ $\theta_k$ and R $R_l$ and m $M_j$ and $b_z$ $Bz_m$ THEN $U_2$ $E_{cv}$
Rule 3: $T_{cv}$: IF $\theta$ $\theta_k$ and R $R_l$ and m $M_j$ and $b_z$ $Bz_m$ THEN $U_3$ $T_{cv}$
(k = 1, 2, 3 . . . 18, l = 1, 2, 3 . . . 20, j = 1 . . . 5, m = 1, 2)

The third step is to transfer the task properties and energy state to the control structure in order to reveal the task and energy relationship. For this, a 1-input, 8-output selection algorithm was developed. The task selection is set as the input. According to the selection of the tasks, battery, constant energy, estimated energy, time, and distance $V_{max}$ and $V_{min}$, feature outputs are selected. The output values of these properties are 0 and 1. The known energy type is defined by the constant energy, and the unknown energy type is defined by the estimated energy. Battery selection is available for both situations. In the

fourth stage, the amount of energy to be used and $V_{fmax}$ can be found according to the task–energy relationship. The whole road is considered as a whole. In the first case, the initial speed $V_{max}$ is assumed. If constant energy usage is selected, the total battery amount to be used is obtained as $E_{tn}$ according to the $E_t$ duty cycle status. According to the known total energy amount of the curved roads $E_{cv}$, the total energy amount of straight paths $E_{st}$ is found. To find $V_{fmax}$ according to the current $E_{st}$ value, acceleration/deceleration accelerations ($a_{ast}/a_{dst}$) and energies of accelerations $E_{ast}/E_{dst}$, distances $X_{ast}/X_{dst}$ and durations $T_{ast}/T_{dst}$ are obtained. This process is performed with two fuzzy structures created according to the acceleration profiles. The structures used for acceleration and deceleration are the same. Let us examine this fuzzy structure with its increasing acceleration structure.

The fuzzy system consists of four inputs with $V_{max}/V_{min}$, $V_1$ (initial speed), $m$, and $b_z$, and four outputs with $a_{ast}$, $E_{ast}$, $X_{ast}$, and $T_{ast}$. Membership functions and rule tables were created according to the intervals and the result values of the acceleration profiles obtained previously.

Fuzzy rule structures:

$a_{ast}$: IF $V_{max}/V_{min}$ $V_i$ and $V$ $V_l$ and $m$ $M_j$ and $b_z$ $Bz_m$ THEN $U_1$ $a_{ast}$, $U_2$ $X_{ast}$, $U_3$ $E_{ast}$, $U_4$ $T_{ast}$
$a_{dst}$: IF $V_{max}/V_{min}$ $V_i$ and $V$ $V_l$ and $m$ $M_j$ and $b_z$ $Bz_m$ THEN $U_1$ $a_{dst}$, $U_2$ $X_{dst}$, $U_3$ $E_{dst}$, $U_4$ $T_{dst}$

Then, zero acceleration and straight path travel energy in accordance with the $V_{max}$ choices of the task are obtained from the $E_{ct}$ acceleration profiles. The fuzzy structure created for this purpose consists of three inputs, with $V_{max}/V_{min}$, $m$, and $b_z$, and the single output $E_{ct}$. At the same time, $X_{ct} = X_t - (X_{ast} + X_{dst})$ and $T_{ct} = T_t - (T_{ast} + T_{dst})$ values are obtained for time and distance controls.

Fuzzy rule structure:

IF $V_{max}/V_{min}$ $V_i$ and $m$ $M_j$ and $b_z$ $Bz_m$ and $X_{ct}$ $X_k$ and $T_{ct}$ $T_l$ THEN $U_1$ $E_{ct}$

The result is an estimated straight path $E_{stv}$ with values $E_{ast} + E_{dst} + E_{ct}$. If $E_{stv} => E_{st}$, then there is no problem and $V_{fmax} = V_{max}$. However, if $E_{stv} > E_{st}$, the $V_{fmax}$ value is reduced to a lower value than that defined in the profiles, to obtain a speed suitable for the current energy. The speed profile range is 0.2 m/s. There is a control structure at this point regarding whether the energy is met or not.

When the estimated energy type is selected, $E_{cv}$ is obtained considering the characteristics of the task, and $E_{st}$ is obtained according to the battery status. $E_{stv} = E_{ast} + E_{dst} + E_{cv}$ is found by performing the same operations as the first energy determination method. However, this time the energy comparison is made with the battery. When $E_{stv} > B_t$, the $V_{fmax}$ value is recalculated. Unlike other energy types, if time or distance is not selected, the state parameter is added. Condition parameters are criteria such as the estimated distance or time, or a constant speed for a given area or charging point. It is necessary to adjust its remaining energy and speed according to these criteria. As a result, $V_{fmax}$, $V_{cv}$ values suitable for the energy state are obtained.

With the processes so far, general speed and acceleration values are found in accordance with the energy state. However, this general acceleration value may not be suitable for every curve speed and straight road segment. To remedy this situation, acceleration and deceleration accelerations must be found for each curve and straight path. Thus, smoother speed transitions and greater energy savings can be achieved. For this, load, ground friction, $V_{fmax}$, and $V_{cvi}$ are used. Each $V_{cvi}$ and $V_{fmax}$ value is obtained as $a_{asti}$ and $a_{dsti}$ by entering the acceleration profile selection structure in the first operation. The speed of the i'th curve is $V_{cvi}$, the incremental acceleration is $a_{asti}$, and the deceleration acceleration is $a_{dsti}$. The final stage is the establishment of energy balance. Energy consumption may also change according to the changing load, ground, and acceleration conditions at each straight and curved road crossing. In this case, if there is a significant difference between the energy consumed for the remaining section of the road and the energy to be used, the energy must be distributed in a balanced way. To do this, a solution was produced with the Gini coefficient. If the Gini coefficient is more than 0.2, the system is operated in accordance

with the state of the energy again, as in the first case. New movement values are obtained for the remaining energy. In repetitive structures, the stability of the task is ensured by the redistribution of energy. The SIMULINK model of the proposed model is shown in Figure 8.

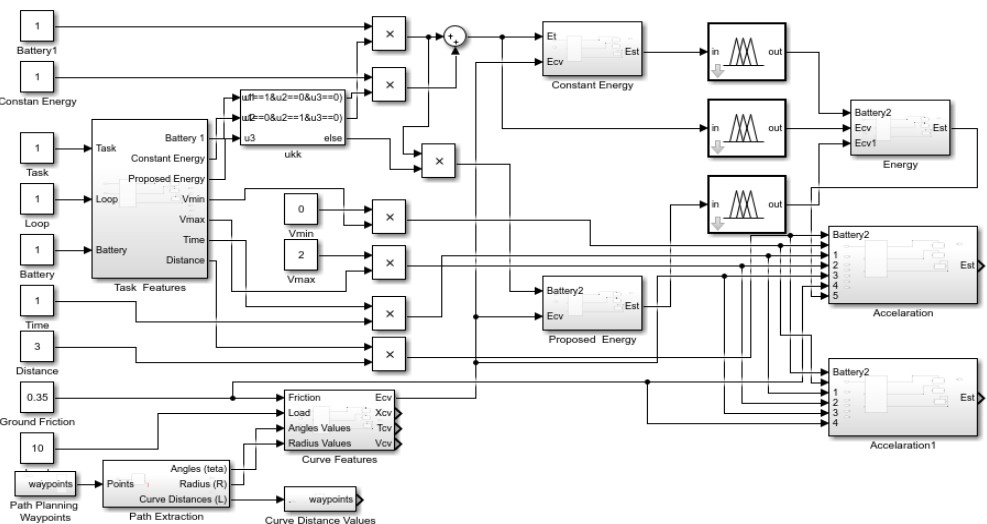

**Figure 8.** Speed–acceleration MATLAB/SIMULINK fuzzy control structure.

### 2.9. Trapeze Speed Profile

An important point in the position control of mobile robots is the creation of optimal speed profiles based on fixed distances or times to obtain the desired position response. To achieve this, many speed profiles, such as the trapezoidal speed profile, the triangular speed profile, the S-curve and sinusoidal speed profiles, and minimum-energy-centered speed profiles have been developed. Among these, the trapezoidal speed profile is one of the more popular profiles due to its simple structure and fast response times.

The trapezoidal speed profile consists of three parts: constant speed, acceleration, and deceleration zones for a given road section. Acceleration and deceleration are usually performed with a certain constant acceleration. Acceleration always tends to reach a maximum speed. This method considers the path as a straight line. It does not offer a solution for the curves of the road. Due to this feature, it is disadvantageous in terms of deviation from the road and energy efficiency. Nonetheless, it has been used in separate speed profile studies for straight and curved roads. However, in these studies, it was seen that sudden acceleration, sudden deceleration, and oscillations occurred at the intersection of curved roads. In some cases, the acceleration profile can be triangular, depending on the length of the target distance of the road. Figure 9 shows the trapezoidal and triangular acceleration profiles.

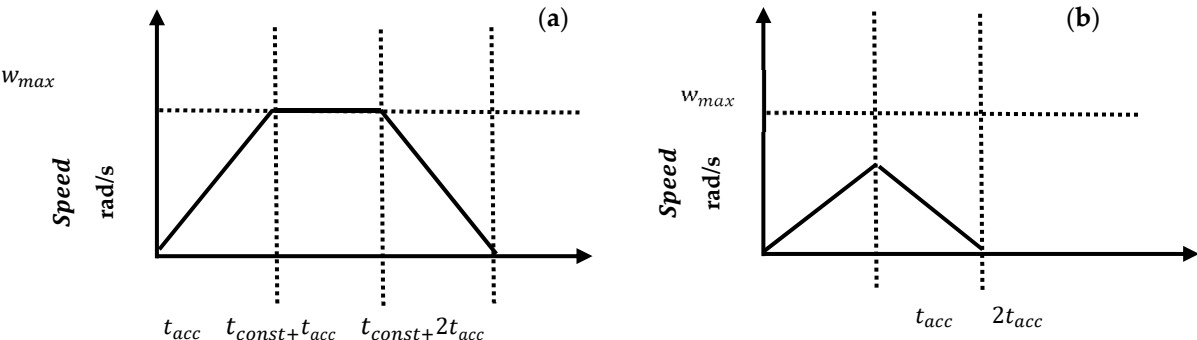

**Figure 9.** Speed profiles: (**a**) trapezoidal speed profile, (**b**) triangular speed profile.

### 3. Simulation Application of the Proposed Speed—Acceleration Control Model

A simulation environment was created in MATLAB/SIMULINK to measure the performance of the developed energy-efficient speed-acceleration control model. The responses of the developed control structure to time, energy, distance, and environment dynamics were examined. The values obtained in the proposed speed–acceleration control structure were compared with the trapezoidal speed profile with constant acceleration and a maximum speed.

The parameters used in the simulation were taken from the real environment information of the experimental study in Figure 10. The parameters of the mobile robot are $K_t = 0.8693$ N-m/A, $K_e = 0.8883$ V/rad/s, $J_m = 0.038$ Kg/m$^2$, $b_m = 0.00075$ N-m/rad/s, $R_a = 53.94$ Ohm, and $L_a = 1$ mH ve $b_z = 0.7$ Nm/rad/s. (s: second, N-m: Newton-Meter.)

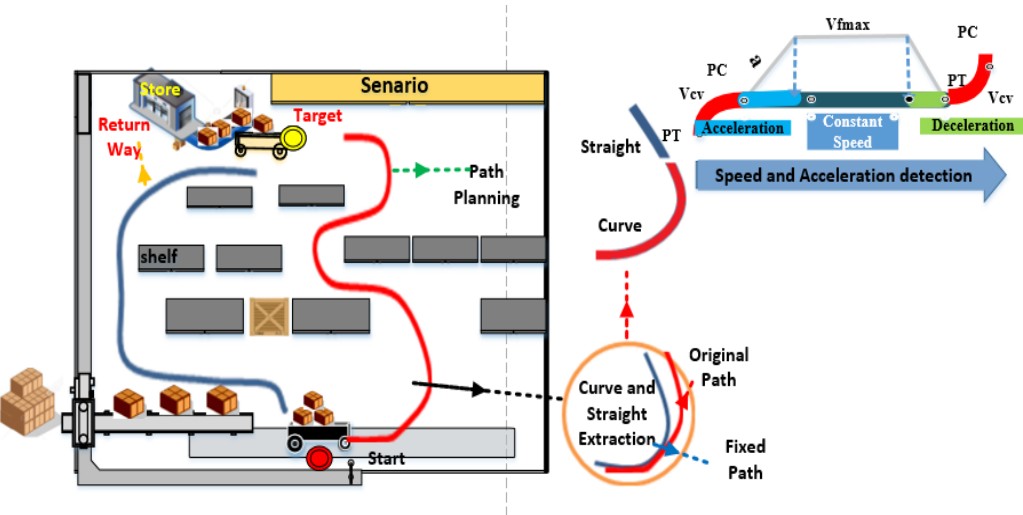

**Figure 10.** Speed acceleration application motion and operations.

The transfer function obtained according to these parameters is as follows [30,31]:

$$G_s = \frac{\omega(s)}{V(s)} = \frac{0.8693}{(1.s + 53.94)(0.038.s + 0.00075) + 0.8693^2} \qquad (29)$$

The Artificial Potential Field algorithm was used for path planning and the Pure Pursuit algorithm was used for path tracking. However, the speed and acceleration profile selection was carried out using the model developed in this study. The required parameters for the Pure Pursuit algorithm were determined as desired linear speed = ~1.5 (m/s), look ahead = 0.3 (m), and angular speed = ~20 (rad/s). PID was used for speed control. Its parameters are $K_p = 22.5$, $K_i = 1.75$, and $K_d = 10$. The ground was chosen as stone and soil. Their coulomb friction coefficients are $\mu_{st} = 0.5$ and $\mu_{sg} = 0.85$, for static, and $\mu_{kst} = 0.4$, and $\mu_{ksa} = 0.5$ for dynamic. The speed increase parameter is cs = 20 [32–34]. As is known, energy measurement consists of sensor, control, and motion parts. In this study, the power consumed by the sensor was taken as $P_{sensor} = 1.2$ watts, $P_{control} = P_{standby} = 1.4$ watts and $P_{startup} = 2.4$ watts for the control card. Other parameters required for energy consumption are $\phi = 0.7$, $\epsilon = 0.01$, $\sigma = 0.001$, and $\lambda = 0.1$ [35].

### 3.1. Simulation Scenario

It is assumed that the transport robots are in a factory environment where there are no dynamic obstacles; they unload the incoming load to the specified warehouse and then return to the starting point. The unloading time of the mobile robot to the warehouse is neglected. The assumed parameters are: ground stone, an initial load of zero, movement speed Vmax, and full battery B(t) = 5000 J. The scenario area is 5 × 5 m in size, as shown in

Figure 11a. Initially, the paths between the starting point and the warehouse are as shown in Figure 11b. The routes for leaving and returning are considered to be the same.

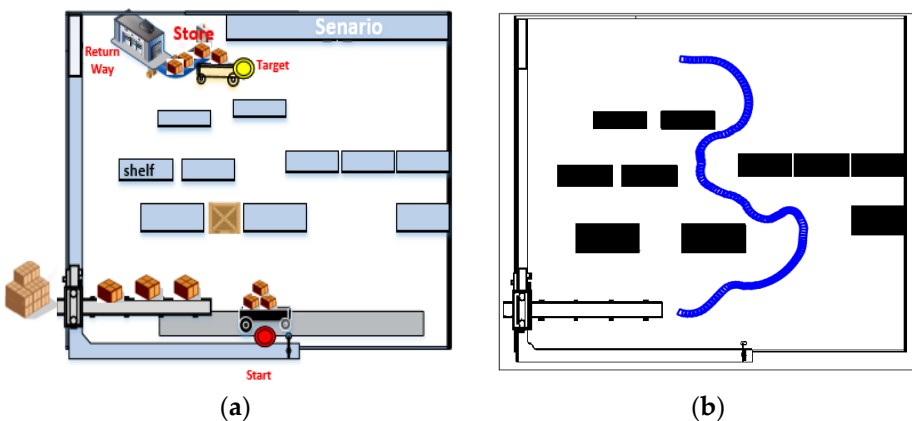

(**a**)        (**b**)

**Figure 11.** (**a**,**b**) Task and path planning of the mobile robot.

### 3.2. Path Extraction

Then there is the matter of the curve and straight path inferences. The curve threshold value for curves was taken as 1.25. As shown in Figure 12, five curved and four straight roads were obtained.

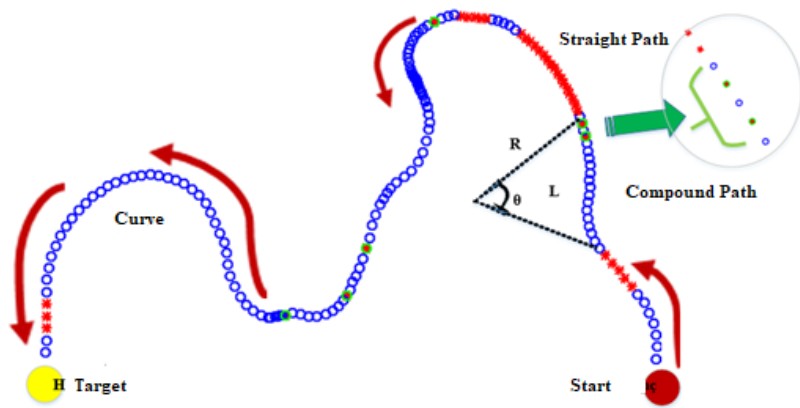

**Figure 12.** Detection and smoothing of segments of the road.

Based on the curvature R, θ, $\mu_{kst}$, and m information of the formed smoothed path, and $V_{cv}$, $E_{cv}$, $T_{cv}$ ve $X_{cv}$, and ve $X_{st}$ information from the existing fuzzy structure were obtained from the path extraction algorithm, as shown in Table 1.

**Table 1.** Path information.

| | | | Fuzzy Model | | | | | Path Inferences | |
|---|---|---|---|---|---|---|---|---|---|
| | | Inputs | | | | Outputs | | | |
| | R (m) | θ (rad) | $\mu_{kst}$ | M (Kg) | $V_{cv}$ (m/s) | $E_{cv}$ (J) | $T_{cv}$ (sec) | $X_{cv}$ (m) | $X_{st}$ (m) |
| 1 | 3.005 | 57.01 | 0.7 | 10 | 0.554 | 372 | 5.3 | 2.99 | 2.927 |
| 2 | 5.221 | 60.23 | 0.7 | 10 | 1.152 | 422 | 4.8 | 5.6 | 5.197 |
| 3 | 3.767 | 21.52 | 0.7 | 10 | 0.605 | 365 | 3.06 | 1.415 | 2.206 |
| 4 | 11.919 | 175.2 | 0.7 | 10 | 1.085 | 1380 | 18.8 | 18.63 | 2.0 |
| 5 | 5.012 | 5.71 | 0.7 | 10 | 0.715 | 602 | 0.69 | 0.5 | - |

## 4. Simulation Results

Five sub-scenarios were created to reveal the energy efficiency, mission stability, durability, and response to load and ground changes provided by the proposed speed–acceleration control model.

Scenario 1: Energy Efficient Speed–Acceleration Profile. Energy 5000 J and distance 40 m, duration = 50 sec. For the trapezoidal speed profile with a maximum speed = 1.5 m/s, the acceleration for each straight road is a=0.2 m/s$^2$.

As presented in Tables 2 and 3, and Figures 13 and 14, the velocity–acceleration profiles produced by the proposed model provide less energy and less deviation from the road, and travel with a maximum speed closer to the curve speeds. Although the trapezoidal velocity profile provides less travel time, it can be observed that increasing deviations from the road increases energy consumption and reduces safe travel. At the same time, unlike in the trapezoidal speed profile, task, energy, and motion compatibility were realized in the proposed model.

**Table 2.** Task motion analysis with the proposed method.

|  | Time (sec) | Distance (m) | Vmax/Vfmax (m/s) | ~Energy Consumption (J) | Task Performance |
|---|---|---|---|---|---|
| Proposed | 50 | 41.46 | ~1.2 | 4373 | Task completed |
| Trapeze | 44 | 45.22 | 1.5 | 4730 | Task completed |

**Table 3.** Task motion analysis with the proposed method.

| Proposed | | | | | Trapeze | | | |
|---|---|---|---|---|---|---|---|---|
| $V_{cv}$ (m/s$^2$) | $E_{cv}$ (J) | $E_{st}$ (J) | $a_{ast}$ (m/s$^2$) | $a_{dst}$ (m/s$^2$) | $V_{cv}$ (m/s$^2$) | $E_{cv}$ (J) | $E_{st}$ (J) | $a_{ast}/a_{dst}$ (m/s$^2$) |
| 0.554 | 372 | 280 | 0.245 | 0.256 |  | 415 | 308 |  |
| 1.152 | 422 | 354 | 0.315 | 0.342 |  | 468 | 397 |  |
| 0.605 | 365 | 352 | 0.245 | 0.28 | 1.5 | 403 | 384 | 0.2 |
| 1.085 | 1380 | 246 | 0.254 | 0.32 |  | 1451 | 262 |  |
| 0.715 | 602 | - | - | - |  | 642 |  |  |

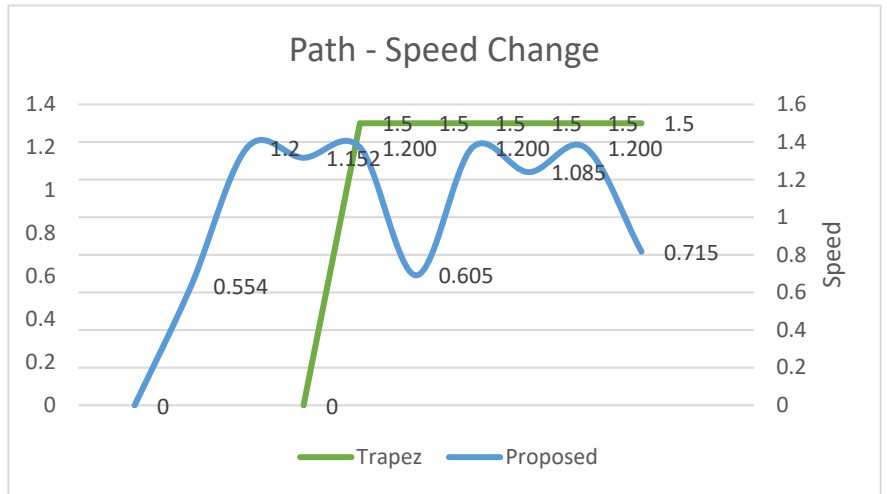

**Figure 13.** Proposed and Trapeze models: curved–straight path segment speed change.

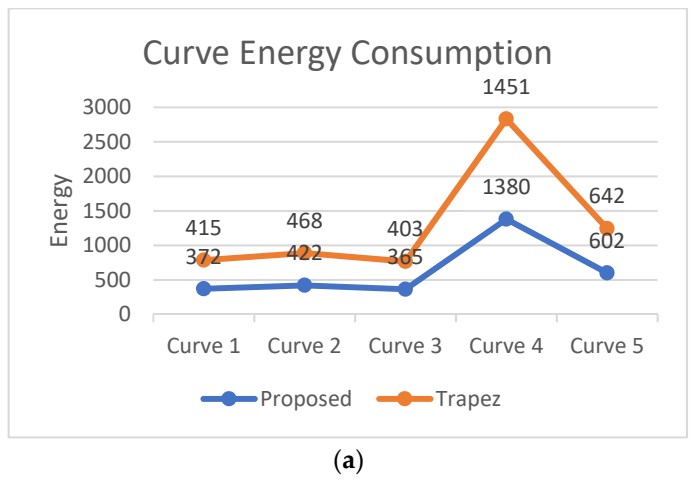
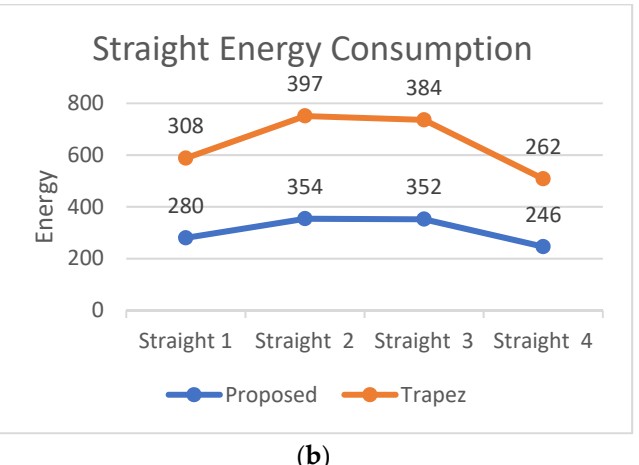

(**a**)  (**b**)

**Figure 14.** Proposed and Trapeze speed profiles: (**a**) curve energy consumption: Vmax = 1.2, (**b**) straight energy consumption: Vmax = 1.5.

Scenario 2: Energy-Balanced Travel using the Gini Coefficient. The results obtained by the balanced distribution of the remaining energy by the Gini coefficient, according to the change in the estimated amount of energy consumption in dynamic environments, are presented in Table 4. This method appears to achieve a new maximum velocity relative to the remaining energy. (Note that 100 J unexpected energy consumption is added after the first curve path and the first straight path.)

**Table 4.** Energy-balanced travel based on the Gini coefficient.

| Static Environment | | | Dynamic Environment | | |
|---|---|---|---|---|---|
| Average Gini Coefficient | $V_{fmax}$ (m/s) | Total $E_{st}$ (J) | Average Gini Coefficient | $V_{fmax}$ (m/s) | Total $E_{st}$ (J) |
| ~0.12 | ~1.18 | 1232 | 0.254 | ~1.13 | 1098 |

Scenario 3: Effect of Ground Friction and Load Change. The application, which was carried out on stone ground, was repeated on earthy ground. Table 5 shows the changes.

**Table 5.** Effect of ground friction.

| Stone Ground | | | | | | Soil Ground | | | | |
|---|---|---|---|---|---|---|---|---|---|---|
| $V_{cv}$ (m/s) | $E_{cv}$ (J) | $E_{st}$ (J) | Total Energy (J) | $a_{ast}$ (m/s$^2$) | $a_{dst}$ (m/s$^2$) | $E_{cv}$ (J) | $E_{st}$ (J) | Total Energy (J) | $a_{ast}$ (m/s$^2$) | $a_{dst}$ (m/s$^2$) |
| 0.554 | 372 | 280 | | 0.245 | 0.256 | 383 | 297 | | 0.251 | 0.258 |
| 1.152 | 422 | 354 | 4373 | 0.315 | 0.342 | 428 | 359 | 4460 | 0.318 | 0.346 |
| 0.605 | 365 | 352 | | 0.245 | 0.28 | 371 | 357 | | 0.249 | 0.297 |
| 1.085 | 1380 | 246 | | 0.254 | 0.32 | 1393 | 251 | | 0.259 | 0.334 |
| 0.715 | 602 | - | | - | - | 621 | - | | - | - |

The load amount was changed to 50 kg and the application was repeated using the same acceleration values. Table 6 and Figure 15 show the changes. It can be seen that the load has an average of an 8% effect on the energy consumption, while the ground friction has an effect of 3%.

**Table 6.** Effect of load change.

| | 10 Kg | | | | 50 Kg | |
|---|---|---|---|---|---|---|
| $V_{cv}$ (m/s) | $E_{cv}$ (J) | $E_{st}$ (J) | $a_{ast}$ (m/s$^2$) | $a_{dst}$ (m/s$^2$) | $E_{cv}$ (J) | $E_{st}$ (J) |
| 0.554 | 372 | 280 | 0.245 | 0.256 | 431 | 311 |
| 1.152 | 422 | 354 | 0.315 | 0.342 | 492 | 388 |
| 0.605 | 365 | 352 | 0.245 | 0.28 | 403 | 387 |

**Table 6.** *Cont.*

| 10 Kg | | | | 50 Kg | | |
|---|---|---|---|---|---|---|
| 1.085 | 1380 | 246 | 0.254 | 0.32 | 1412 | 273 |
| 0.715 | 602 | - | - | - | 658 | |

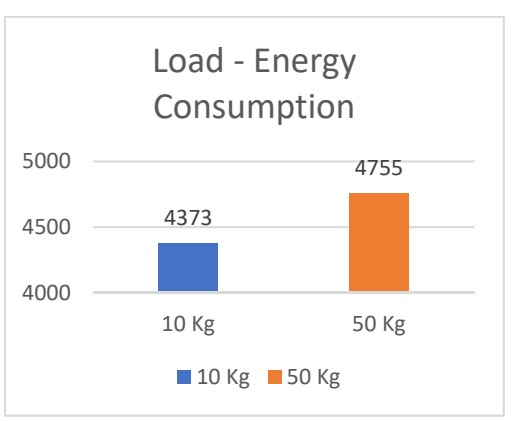 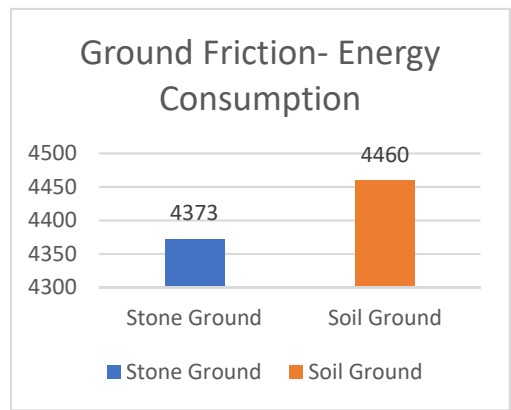

**Figure 15.** Effect of load and ground changes on energy consumption.

Scenario 4: Insufficient Energy Travel Stability. The total estimated energy consumption in the example application is 3000 J. The stability of the proposed control structure for the completion of the task in cases where the energy is not sufficient is shown in Table 7 and Figure 16. As can be seen in the table, when the proposed control structure does not have enough energy, it ensures the stability of the task by providing energy and speed balance.

**Table 7.** Insufficient energy travel stability.

| | Time (sec) | Vmax/Vfmax (m/s) | ~Energy Consumption (J) | Task Performance |
|---|---|---|---|---|
| Sufficient | 50 | ~1.2 | 4373 | Completed |
| Insufficient | 68 | 0.92 | 2857 | Completed |

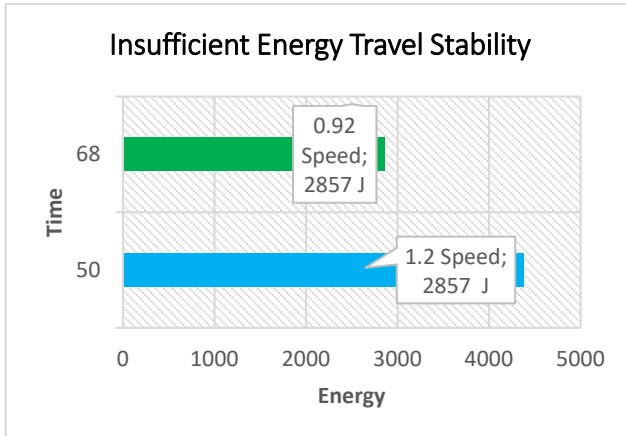 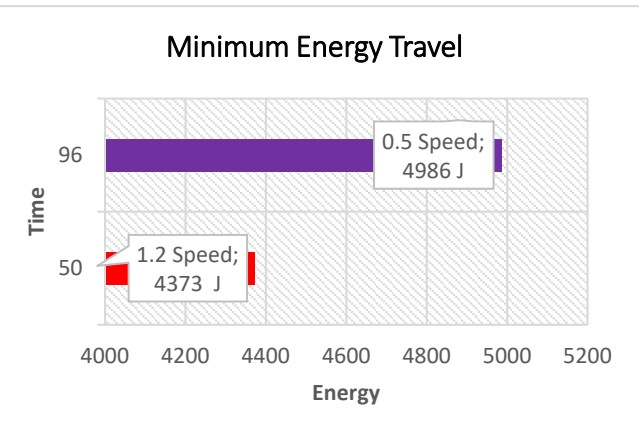

**Figure 16.** Travel energy consumption levels with little energy and minimal energy.

Scenario 5: Minimum Energy Travel. This is the type of travel expected in missions where more scanning of the area is performed. The robot is expected to travel to the targeted point with the available energy with minimum acceleration over a long time. In this case, the minimum acceleration for the straight road can be determined. However, the curve speeds can be greater than the maximum speed. This situation brings with it sudden speed

changes and excessive energy consumption. In order to prevent this, the maximum speed limit is imposed on the curve speeds. The accelerations are the same. The travel results obtained according to the maximum speed limit are shown in Table 8.

**Table 8.** Minimum energy travel.

| Time (sec) | Vmax/Vfmax (m/s) | ~Energy Cosumption (J) | Task Performance |
|---|---|---|---|
| 50 | ~1.2 | 4373 | Completed |
| 96 | 0.5 | 4986 | Completed |

### 5. Conclusions

As a result, it can be observed that the speed–acceleration control structure produces smooth paths and speed–acceleration profiles suitable for the structure of the roads and for the mobile robot. The energy balance, realized with the Gini coefficient, ensures the appropriate motion planning of the system. Thus, it is ensured that the tasks show optimum energy stability and endurance. At the same time, the developed method's ability to respond online to changing environmental dynamics gives it a modular feature. Determining the speed and acceleration profiles according to the curve speeds of the road enables the mobile robot to follow the reference path in the best way, to prevent excessive energy consumption due to unnecessary acceleration, and to avoid sudden speed increases and deceleration compared to the reference path. The negative consequences of the mobile robot traveling the entire path at maximum speed are prevented by the maximum speed adjustment, which is recreated according to the energy and speed balance.

**Author Contributions:** Conceptualization, G.G. and İ.T.; methodology, G.G.; software, İ.T.; validation, GG. and İ.T.; formal analysis, G.G.; investigation, G.G.; resources, G.G.; data curation, G.G and İ.T.; writing—original draft preparation, G.G.; writing—review and editing, G.G. and İ.T.; visualization, G.G.; supervision, İ.T.; project administration, İ.T.; funding acquisition, G.G. All authors have read and agreed to the published version of the manuscript.

**Funding:** This research received no external funding.

**Conflicts of Interest:** The authors declare no conflict of interest.

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
