# Peer review of "A Novel, Energy-Efficient Smart Speed Adaptation Based on the Gini Coefficient in Autonomous Mobile Robots"

_electronics, doi:10.3390/electronics11192982_

Round 1

Reviewer 1 Report

The authors focus their study on the field of mobile robots by supporting their autonomous operation in an energy efficient manner. The main goal of the authors is to distribute the energy in a balanced way considering the different qualifications of each task of the autonomous mobile robot aiming at guaranteeing the movement of the robots at their optimum speed. Towards this direction, the authors introduce an online intelligent speed and acceleration adaptation method that exploits the task structure and the energy balance is achieved for a specific path that overcomes the shortcomings of the existing models. The manuscript is overall well written and easy to follow and the authors have well thought out their main contributions. The provided theoretical analysis is concrete, complete, and correct and the authors have provided all the intermediate steps in order to enable the average reader  to easily follow it. The authors are highly encouraged to address the following comments in order to improve the scientific depth of their manuscript, as well as the quality of its presentation. Initially, in Section 1, the provided related work needs to be substantially rewritten in order to be presented by using more summative language in order to better identify the research contributions that have already been performed in the literature and the research gap that the authors try to address. Specifically, in Section 1, the authors need to discuss existing low computational complexity approaches, such as A UAV-enabled Dynamic Multi-Target Tracking and Sensing Framework, doi: 10.1109/GLOBECOM42002.2020.9322567, that exploit intelligent algorithms in order to perform the path planning of autonomous robots. In Section 4, the authors need to include an additional subsection discussing the implementation cost and the computational complexity of the proposed framework in order to be implemented in a realistic environment. Based on the previous comment in Section 4, the authors need to include some additional numerical results quantifying the computational complexity of the proposed framework. Finally, the overall manuscript needs to be checked for typos, syntax, and grammar errors in order to improve the quality of its presentation.

Author Response

RESPOND TO REVIEWER

Reviewer 1:

The authors focus their study on the field of mobile robots by supporting their autonomous operation in an energy efficient manner. The main goal of the authors is to distribute the energy in a balanced way considering the different qualifications of each task of the autonomous mobile robot aiming at guaranteeing the movement of the robots at their optimum speed. Towards this direction, the authors introduce an online intelligent speed and acceleration adaptation method that exploits the task structure and the energy balance is achieved for a specific path that overcomes the shortcomings of the existing models. The manuscript is overall well written and easy to follow and the authors have well thought out their main contributions. The provided theoretical analysis is concrete, complete, and correct and the authors have provided all the intermediate steps in order to enable the average reader  to easily follow it. The authors are highly encouraged to address the following comments in order to improve the scientific depth of their manuscript, as well as the quality of its presentation. Initially,

Concern #1:  

in Section 1, the provided related work needs to be substantially rewritten in order to be presented by using more summative language in order to better identify the research contributions that have already been performed in the literature and the research gap that the authors try to address.

Specifically, in Section 1, the authors need to discuss existing low computational complexity approaches, such as A UAV-enabled Dynamic Multi-Target Tracking and Sensing Framework, doi: 10.1109/GLOBECOM42002.2020.9322567, that exploit intelligent algorithms in order to perform the path planning of autonomous robots.

Author Respond #1:

Thank you for your valuable comment. The introduction part has been rewritten in line with your suggestions.

Updated text:

Mobile robots have limited energy resources. This situation has revealed the efficient use of energy as an effective solution.At this point, many energy efficient task and motion planning (short path planning, path smoothing, velocity profiles, etc.) strategies have been developed. In both types of studies, the focus is on reducing the motor running time and the power it consumes during travel in the mobile robot system. Because the most consumption in the system takes place in the engines. In terms of energy efficiency, task and short path studies are not sufficient solutions on their own. It is also necessary to consider the power consumption during travel. The power consumption of the engines during travel relies heavily on correct steering angle, speed and acceleration. In mission and motion planning studies, energy efficient speed optimizations are neglected in terms of task durability and stability, or they are not taken into focus as an effective parameter.

Current energy-efficient speed profile studies generally take into account the structure of the road, but not the structure , durability, and stability of the task. Energy efficient speed profiles are based on distance and time. Straight and curved segments of the road have an important place in determining the speed. Initially, a trapezoidal velocity profile was used in applications. However, Wang et al. revealed that this method is not optimal in terms of total energy. Kim and Kim have separately presented energy-efficient velocity profiles based on constant travel time for straight paths and rotational trajectory. However, a speed profile combining both road sections has not been established. Tokekar et al., on the other hand, divided the road into segments and presented energy-efficient combined optimum speed profiles for straight and curved roads. Optimum velocity profiles are calibrated according to the specific load and ground according to the polynomial model. Calibration is done offline. Recalibration is required for each ground and load. Tokekar et al. In another study stated that the length of the return path increases if the maximum speed is not limited. The first study considering the relationship between energy amount and time was carried out by Broderic et al. They divided the road into straight and turning segments and provided acceleration according to the offline determined gain parameters for each segment of the road. But they could not achieve the balance of time, energy and speed.

The energy efficiency in these studies is a result of the velocity profile used. In these studies, the amount of energy use, its limit or the criteria by which energy saving is determined is not available. The amount of energy remaining in the battery was not taken into account at the point of completion of the mission. However, mobile robots perform their movements according to certain tasks and the amount of power in the battery. The definition of the task and the energy relationship in the battery affect the amount of energy to be used, the endurance of the task, the speed and acceleration profile. Otherwise, only an energy efficient operation will be put forward. At the same time, there is no limit to the maximum speed in terms of energy use in the current studies. This can lead to the deterioration of the energy, speed and task relationship. At the same time, moving at maximum speed, as in the Broderic and Sharallimo studies, will unbalance the distribution of energy between segments of the path to meet the criteria of the task [9-16]. There is no measure to correct this imbalance in current studies. In addition to these, acceleration was generally chosen as constant in all these studies. However, it has been seen in real car studies that creating acceleration and acceleration/deceleration distance according to curved roads or the point where the speed change will occur provides energy saving and safe travel. A dynamic acceleration profile for online load changes is also often not available.

As understood from the studies, it is necessary to consider the task description, battery status, amount of energy used and motion dynamics (maximum speed limit, acceleration, load and ground, etc.) as a whole in order to create the optimum energy efficient speed profile. Otherwise, energy balance and optimum speed cannot be achieved as in the current speed profiles.

The remaining parts of the study; In the second part, the methods used are mentioned. In the third chapter, the integrated structure of control and motion planning with these methods is revealed step by step. In the fourth part, simulation studies were carried out and comparison with the trapezoidal velocity profile was performed.

Concern #2 Finally, the overall manuscript needs to be checked for typos, syntax, and grammar errors in order to improve the quality of its presentation.

Author Respond #2: Article spelling errors have been corrected in line with your suggestions.                                  

Concern #3:  In Section 4, the authors need to include an additional subsection discussing the implementation cost and the computational complexity of the proposed framework in order to be implemented in a realistic environment. Based on the previous comment in Section 4, the authors need to include some additional numerical results quantifying the computational complexity of the proposed framework.

Author Respond #3:

Since the cost function and complexity of the developed control structure could not be developed sufficiently, it was not added. can be developed in sufficient time.

Author Respond #4:

In addition, the results sections have been made more understandable.

The shapes have been redrawn.

The operability of the acceleration and control structure has been revealed more clearly.

Reviewer 2 Report

The article is a relevant subject and written at a good level. 

However, there are some points that need to be corrected, so I recommend a major revision of the article.

The title of the article completely corresponds to it.

However, the article has a few points that need to be corrected:

In Figure 1, you need to redo the labels because they cover part of the arrows.

In the text of the article, there are many errors, missing letters, periods, etc.

All equations have bad formatting. This should be fixed.

Figures 4 and 5 should be revised to ensure better quality.

How is the robot speed controlled and accelerated? How does the speed and acceleration control system work when there are differences between the expected speed indicators and the real ones? (This can occur when driving on different surfaces with different coefficients of friction between the wheels and the floor.)

The authors indicate that they use a trapezoidal speed profile of the robot. However, in real life such a profile can be achieved only with a straight trajectory. In the case of a mobile robot and a simulated trajectory, centrifugal forces and an increased coefficient of friction on turns will act on the robot. This will lead to a decrease in speed and deviation from the trajectory. The author should comment on it or take it into account in his research.

The biggest shortcoming of the article is the presentation of the results. All the results are presented in tables and there are so many of them that it is very difficult to compare the results. The authors need to revise the presentation of the results and provide graphical dependences of energy, speed and other parameters responsible for the movement of the robot.

The conclusions should be clearer with the results obtained, how much energy use is reduced with the control algorithm used.

Author Response

In line with your valuable suggestions, explanations about the general arrangement of the article are available in the attached file.

Round 2

Reviewer 1 Report

The authors have addressed in detail the reviewers comments.

Author Response

Thank you for your interest and valuable contribution to the development of the article.

Good Works. Take care of yourselves. Thanks for everything

Reviewer 2 Report

The authors of the article significantly improved the manuscript. However, there are a few more comments, so I recommend a minor revision of the article.

- There are still many errors in the formatting of the article (for example, the units of measurement in the text are indicated differently in one case is sec. in another is s, ).

- Also, sometimes there are symbols not listed in the indexes.

- The authors never presented the obtained results graphically.

- In the article on problems with the names of the sections, section 4 should be conclusions, but for some reason, it is the results.

Author Response

Reviewer 2:

The authors of the article significantly improved the manuscript. However, there are a few more comments, so I recommend a minor revision of the article.

 Concern #1:  

- There are still many errors in the formatting of the article (for example, the units of measurement in the text are indicated differently in one case is sec. in another is s, ).

Concern #2:  Also, sometimes there are symbols not listed in the indexes..

Author Respond #1:

Based on your valuable suggestions, all symbols and units in the article have been checked.

The following typos have been corrected.

  • Line 154:   equation 2.1:    -- >    Equation 1
  • Equation 5: )             sign : sign function

Equation 29:

  • For the sake of unity in the text, the word "speed" has been used instead of "speed" for the concept of speed.
  • In the text, the term "motor" was used instead of the concept of "engine".
  • Line 238 : (( )), Line 305 ( ),),  gibi yazım hataları düzeltildi.
  • Line 159 : Added missing symbol and unit emotes and introductions. Corrected misspellings
  • The static friction and kinetic friction  or

    (5)

    The viscous friction

    (7)

    The Stribeck effect       

    (8)

    The friction model;

    (9)

    Angle sepeed (w), radius (r), speed (v), coefficient of kinetic friction (  and statik friction ( , viscous friction ( ), coefficient of transition from static to kinetic friction ( ),  is the force generated in the motor until it reaches the Fs force required for the movement of the mobile robot.  is he Stribeck effect. The total torque generated in the motor during movement. Simulink model available in Figure 3 [20-23];

  • Line 259 : Fixed unit typos such as m/s2 à m/s2
    • Line 480 : Fixed unit typos such as :
    •  N-m/A,  V/rad/s,  = 0,038 Kg/m2, =0,00075 N-m/rad/s,  mH ve =0,7 Nm/rad/s. (s: second, N-m: Newton-Metre)

      Motor dynamic parameters and units are available as stated above in [20-23] articles.

      • Line 188: Other parameters required for energy consumption are Ï•=0.7, =0.01, σ =0.001, λ=0.1 . Here ϵ and λ are heat time constant and σ is velocity heat constant.

      In article 24 are the heat constants used to calculate the amount of heat released during the motion of the motor.

      • Added missing units in tables
      • Fixed "s" expressions as "sec" in the text.
      • Concern #3:        

        - The authors never presented the obtained results graphically.

        - In the article on problems with the names of the sections, section 4 should be conclusions, but for some reason, it is the results.

        Author Respond 2 #:  

        Graphs were created for the results presented in Chapter 4. Added explanations about these charts. In the simulation section, the applications and results related to the sections were written step by step.

          Apart from these, the missing statements were completed. Changes are marked as Turquoise in the main text.
